# Oral Lichenoid Lesion following Dental Implant Placement and Successful Management with Free Gingival Graft: A Case Report with 10-Year Follow-Up

**DOI:** 10.3390/medicina59122188

**Published:** 2023-12-17

**Authors:** Won-Bae Park, Junghun Moon, Seungil Shin, Ji-Youn Hong

**Affiliations:** 1Private Practice in Periodontics and Implant Dentistry, Seoul 02771, Republic of Korea; wbpdds@naver.com; 2Department of Dentistry, Graduate School, Kyung Hee University, Seoul 02447, Republic of Korea; moonpyzun@hanmail.net; 3Department of Periodontology, Kyung Hee University School of Dentistry, Kyung Hee University Medical Center, Seoul 02447, Republic of Korea; shin.dmd@khu.ac.kr

**Keywords:** free gingival graft, palatal soft tissue, titanium, dental implant, oral lichenoid lesion, oral lichen planus

## Abstract

Titanium and metal alloys are widely used in implants, crowns, and bridges in implant dentistry owing to their biocompatibility. In this case report of a 45-year-old female patient, multiple implants were placed in five different sextants at different time points. Notably, oral lichenoid lesions (OLL) occurred in three sextants following implant placement, strongly suggesting that the dental implants or prostheses were the causative factors for OLL. The lesion was of the reticular type with erythematous surroundings and was symptomatic. Although several conservative treatments, including repeated topical application of corticosteroids, were repeatedly continued, no discernible improvement or alleviation of symptoms was observed. Consequently, surgical excision and replacement of the lesion with a free gingival graft (FGG) harvested from the palatal soft tissue were performed. No clinical symptoms or recurrence of lesions were observed during 10 years of follow-up post-FGG.

## 1. Introduction

Oral lichen planus (OLP) is a chronic inflammatory disorder affecting the buccal mucosa, gingiva, and tongue that causes damage to the epithelium and connective tissues [1]. A common clinical characteristic is the presence of white lines (Wickham’s striae) surrounded by a variety of erythematous lesions that are distributed symmetrically. OLP can be associated with symptoms such as discomfort when exposed to spicy or acidic foods, mucosal roughness or stiffness, and gingival lesions that may exhibit desquamation, bleeding, and ulceration [2]. The pathogenesis of OLP involves T-cell-mediated immune dysregulation in predisposed patients, leading to an increase in tumor necrosis factor-α, interferon-γ, and keratinocyte/T-cell/antigen-presenting cell associations [3]. However, the etiological antigenic changes are still unknown. Factors contributing to OLP include psychological stress, genetic background, systemic association, and thyroid dysfunction [1,3]. Although the malignant potential of OLP is controversial, a few large retrospective studies have reported a higher risk of malignant transformation to squamous cell carcinoma (SCC) in OLP populations [4,5].

A spectrum of oral lichenoid lesions (OLL) is recognized, which are considered variants of OLP or a distinct disease, and include lichenoid contact lesions, lichenoid drug reactions, and lichenoid lesions of graft-versus-host disease [6]. Lichenoid contact lesions of OLL are rare but may arise from dental restorative materials, including amalgam, gold, composites, and various other metals, triggering local type IV or delayed allergic contact hypersensitivity reactions [6,7]. Oral lichenoid contact reactions are most commonly associated with dental amalgam and some studies have reported complete or partial resolution of oral lichenoid contact lesion following the replacement of amalgam restorations with positive patch test [8,9]. It is difficult to distinguish between OLP and OLL because of their similar clinical and histopathological features. Key clinical details for diagnosing lichenoid contact lesions include their geographical relationships with restorations and the absence of specific drug intake indicative of lichenoid drug reactions.

Dental implant placement in patients with OLP was not recommended in the past due to concerns about possible inflammatory trauma and negative effects on epithelial attachment to the titanium surface [10]. A prospective study reported high failure of 42 implants out of 55 within a short loading period of 7 to 11 weeks in an active erosive phase, which suggested that implant surgery should be avoided until the remission of atrophic or erosive lesion was achieved [11]. In addition, it is necessary to exclude dysplasia and SCC by histopathologic assessment before the active treatment and periodically monitor the possible malignant transformation of OLP. However, some clinical studies have suggested that the survival rate of implants in patients with controlled OLP is comparable to that in patients with healthy mucosa [12]. Most studies have evaluated the clinical outcomes of implants placed in areas where the lesions were controlled preoperatively by using topical and/or systemic corticosteroids with various ingredients and protocols.

Cases of OLL developing after implant or prosthesis placement have rarely been reported. A clinical study reported titanium (Ti) allergy with a low prevalence of 0.6% from 1500 implant patients, who showed clinical features including allergic symptoms after implant surgery, de-keratinized hyperplastic lesions of peri-implant soft tissues, unexplained implant failures such as spontaneous rapid exfoliation, history of multiple allergies, etc. [13]. However, the lesion is frequently granulomatous on the soft tissues and persistent pain from the bone may be present, which are rather distinctive from typical oral lichenoid contact lesion. Still, the significance of Ti as a cause of allergic reactions in dental implants is not proven [14]. Another retrospective study investigated the proportions and characteristics of patients who complained of pain and discomfort in oral mucosal lesions following dental treatment; OLP and allergic reaction occurred most commonly after implant treatment among patients [15]. Patients with lichenoid contact reactions reported to have higher prevalence of dental metal allergy and 71% of patients showed regression after removal or replacement of dental restorative materials [16]. In patch-tested subjects, the most frequent allergens were mercury followed by gold and nickel. Previous studies have limitations that the sample size is very small, heterogenous dental materials are included, and they cannot clearly identify the causal agent of the lesions. Additionally, long-term results of surgical approaches, such as free gingival graft (FGG) using palatal soft tissue to replace mucosal lesions, are also lacking.

The purpose of this report was to present a case of OLL occurring in multiple regions within a patient following the placement of an implant prosthesis. In addition, we provide the long-term results of a surgical approach using FGG to manage OLL.

## 2. Case Presentation

A 45-year-old female patient who visited a private practice in periodontics and implant dentistry underwent dental implant treatment following the extraction of teeth due to severe periodontitis or root caries in different sextants at different time points during a follow-up period of 21 years (Figure 1a). The patient was a non-smoker and did not have any specific systemic diseases. No preoperative soft tissue lesions were observed in the mucogingival area. Among the five sextants reconstructed with dental implants, clinical features of OLL manifested in the following three regions as described below. The patient reported no concomitant extraoral lesions other than those in the oral cavity.

### 2.1. Mandibular Right Posterior Region

The second premolar, first and second molars in the right mandible were extracted and reconstructed as a cement-retained porcelain-fused-metal (PFM) three-unit bridge with two HA-coated implants (Steri-Oss^®^ Inc., Yorba Linda, CA, USA) (Figure 1b). The patient complained of soreness upon contact with spicy or hot food and experienced discomfort while brushing in the buccal mucosal area around implants during a visit in 2002 after prosthesis placement. Clinical aspects of white striae and erythematous mucosa were observed, of which reticular type OLL was suspected, and the patient was referred to the Oral Medicine at the University Hospital for specialized management of the lesion. For palliative therapy, topical corticosteroid 0.1% dexamethasone ointment (Peridex, Green Cross Corp., Yongin, Republic of Korea) was repeatedly administered and recall visits were made every 2~3 weeks for 4 months; however, the improvement in symptoms was insignificant, and the lesion recurred continuously. In addition, the mandibular right posterior region had a shallow vestibule and lack of keratinized mucosa around implant prosthesis, which made the patient’s cleansing performance difficult. Therefore, a surgical approach involving keratinized tissue augmentation using FGG was performed, which resulted in the disappearance of OLL until the visit in 2011 (8 years after the first FGG). The patient’s attendance to the maintenance visit after the FGG was irregular, with 16~24 months of recall intervals. Self-performed mechanical plaque control in the interproximal areas of posterior teeth was not adequately done and easily showed bleeding on probing with plaque accumulation and gingival redness. At every visit, thorough whole-mouth scaling and root planing with an ultrasonic scaler (EMS, Nyon, Switzerland) and hand instrument were performed. However, advanced peri-implantitis was observed in both the premolar and second molar implants (Figure 1c), and the width of the grafted mucosa was significantly reduced after 3 months of implant removal. Three SLA surface implants (Implantium^®^, Dentium, Seoul, Republic of Korea) were placed and cement-retained PFM splinted crowns were delivered (Figure 1d), following which OLL recurred around the second molar region where the dimension of keratinized mucosa was almost lost (Figure 2a). White striae with erythematous surrounding in the buccal mucosa was observed. FGG was performed again around the mandibular right posterior implants to augment keratinized tissue around the implant and remove OLL concomitantly. Excision of the overlying mucosa, including the OLL, to leave only the periosteum, was performed during recipient bed preparation (Figure 2b). Epithelialized free gingival tissue measuring 25 × 6 × 1 mm was harvested from the opposite right maxillary palate and stabilized on the periosteum of the recipient bed (Figure 2c,d). An increase in keratinized mucosa with a deepened vestibule and relief of the OLL was observed after 3 months of healing (Figure 2e). To evaluate the pathologic tissue condition after the healing of FGG, intraoral clinical photography was taken under a standardized set and a scoring system with a four-point clinical grade scale (0–3) by Axéll and Henriksen [17] was applied. In short, it was subjectively assessed as grade 0 for no improvement or aggravation, grade 1 for improvement but with extensive erythema and/or symptoms, grade 2 for improvement but with some erythema and no symptoms, and grade 3 for healing with neither erythema nor symptoms. The healing score in the mandibular right region appeared to be grade 3, and a healthy mucosal condition was maintained for up to 10 years of follow-up (Figure 2f). Although the patient still showed irregular compliance for the recall visit, the interval was shortened within a year and professional mechanical plaque control was repeated for supportive maintenance therapy.

### 2.2. Maxillary Left Posterior Region

The second premolar and first molar in the left maxilla were placed with two SLA surface implants (Implantium^®^, Dentium, Seoul, Republic of Korea), and cement-retained PFM splinted crowns were delivered in 2013. OLL developed in the buccal mucosa of both keratinized and non-keratinized zones around the implants after 4 months of follow-up, exhibiting clinical features of Wickham’s striae and slight epithelial desquamation (Figure 3a). Oral steroid prednisolone (Solondo 5 mg) was administered 3 times a day for the first 2 weeks, and a mouth rinse of 10 mL of 0.05% dexamethasone gargle was used concomitantly. The painful lesion persisted despite the topical and systemic corticosteroid therapy. In addition, high frenal attachment and a movable mucosal margin around dental implants facilitated plaque accumulation and gingival inflammation. To correct the peri-implant mucosal anatomic conditions and excision of OLL, surgical intervention using FGG was performed. The overlying soft-tissue lesion was excised, leaving only the periosteum (Figure 3b), and the epithelialized free gingival tissue harvested from the same left maxillary palate (18 × 7 × 1 mm) was stabilized on the recipient bed (Figure 3c). The excised soft tissue lesion was fixed in 10% formalin solution and prepared for the histopathological examination. The findings of the histological section revealed separation of squamous epithelium from the underlying connective tissue with degeneration of the basal cell layer and basement membrane, and numerous inflammatory cells were infiltrated predominantly by lymphocytes, suggesting OLL (Figure 3d). The healing score was grade 3 and there was no recurrence of the lesion observed at 6 months (Figure 3e) or 9 years of follow-up (Figure 3f) after prosthetic loading.

### 2.3. Mandibular Left Posterior Region

SLA surface implants (Implantium^®^, Dentium, Seoul, Republic of Korea) were placed in the mandibular left first molar, second molar, and second premolar in 2012, 2014, and 2019, respectively (Figure 1d–f). A cement-retained resin-faced single gold crown was delivered at the first molar implant, and a PFM single crown was made for each second molar and premolar implant. The clinical features of Wickham’s striae and patient’s discomforts were first observed adjacent to the first molar implant during the follow-up for 4 months of prosthesis placement. The extent of lesions in the buccal mucosa gradually increased from the premolar to the ascending ramus with serial implantations (Figure 4a). Similar to the other two regions described above, the mandibular posterior region showed no specific improvement in symptoms despite topical and systemic corticosteroid therapies, and the patient felt difficulty in brushing around the implant due to the movable non-keratinized mucosa. The buccal mucosal lesion was replaced with FGG harvested from the palatal tissue according to the surgical procedure as described above, including excision of the soft tissue (Figure 4b), harvesting of the epithelialized palatal tissue (25 × 7 × 1 mm) (Figure 4c,d), and stabilizing the graft on the recipient bed (Figure 4e). The excised specimen was histologically diagnosed as an OLL, which revealed hyperparakeratinized squamous epithelium, degeneration of the basal cell layer, dissolution of the basement membrane, and dense inflammatory cell infiltration, predominantly by lymphocytes, in the lamina propria beneath the basal cell layer (Figure 4f). The healing score was grade 3, and increased width of the keratinized mucosa in healthy condition around the implants was shown at 6 and 12 months of follow-up (Figure 4g,h).

## 3. Discussion

The present case report demonstrated multiple areas that exhibited OLL following the placement of implant prosthesis in a patient. These lesions resembled the reticular type of OLP, displaying Wickham’s striae and a surrounding erythematous area that caused pain and discomfort when consuming hot, salty, and spicy foods. Histologic examination of biopsies further confirmed the resemblance to OLP, including parakeratinized squamous epithelium separated from the underlying connective tissue, degenerated basal cell layer, and basement membrane. Dense infiltration of inflammatory cells, predominantly lymphocytes, was observed in the lamina propria of the underlying connective tissue. As all lesions exclusively manifested in the buccal mucosa, where the soft tissue was previously healthy, and an OLP-like appearance was observed following the placement of implant prosthesis, a lichenoid contact reaction was presumed to be involved. There were no specific systemic diseases or medication-related issues.

Lichenoid contact reactions are rare; however, some studies have reported contact allergic hypersensitivity to dental amalgam [18]. This process is explained by a host cell-mediated response to the low-level exposure of corrosion products from amalgam, which penetrate the epithelial layer and adhere to keratinocyte surface proteins. Titanium (Ti) and metal alloys are widely used for implants, crowns, bridges, and orthodontic appliances. Ti plays an important role in facilitating osseointegration and provides reliable results in terms of survival rate and biocompatibility. Although the majority of Ti implant failures are related to biological complications, such as biofilm-associated peri-implantitis, another biological issue of allergic reaction to Ti has been suggested [13,14,19]. Data on this issue are scarce; however, some studies have reported the presence of metal particles in the histopathological analysis of peri-implant lesions as a possible consequence of tribocorrosion. Hypothetically, corrosive products from Ti implants or metal alloys from prostheses may be associated with OLLs in susceptible patients. Besides the allergic stimuli from dental material itself, chronic irritation induced by sharp edges, rough surfaces and hard-to-reach sites for oral cleansing, which increase bacterial plaque accumulation, are also suggested to be involved with OLL [16,20].

The main aim of OLL/OLP treatment is to control painful symptoms if present, considering its cyclic nature of remission and exacerbation, and the absence of a permanent cure [21]. Lichenoid contact lesions may occasionally be resolved by removing and replacing suspicious restorative materials; however, complete resolution of the lesion is unpredictable. Risks that may exacerbate the lesion still remain, including potential iatrogenic damage during dental procedures and possible allergic reactions to new restorations [7,21]. Furthermore, the patient in the present case report showed oral lichenoid contact lesions adjacent to both PFM and gold restorations, which limited the alternative choices of dental materials. Topical corticosteroids are considered the first-line treatment due to their fewer adverse effects compared to those of systemic agents. Ointments and suspensions of corticosteroid agents, including 0.1% triamcinolone acetonide, betamethasone, potent fluoridated corticosteroids, and 0.05% clobetasol propionate, are widely used [12,21]. However, the optimal dose, application duration, safety, and efficacy of each agent remain unclear, and no consensus has been reached regarding the treatment guidelines. Unfortunately, in our case, no significant improvements in patient discomfort or clinical features were observed despite continuous topical corticosteroid administration. Consequently, surgical intervention was deemed necessary to excise the soft tissue lesion and graft it with free palatal soft tissue.

In general, surgical excision has not been sufficiently validated as the potential for inflammatory conditions to recur exists. However, a few reports have demonstrated a stabilized gingival condition without OLL recurrence after surgical intervention using free palatal soft tissue, platelet-rich fibrin, or an acellular dermal matrix [22,23]. Notably, FGG using palatal soft tissue is a technique with excellent predictability and is widely used to increase the width of keratinized mucosa around teeth or implants [24,25]. Augmentation of keratinized mucosa around implants can result in more favorable peri-implant health by facilitating oral hygiene maintenance, reducing bleeding scores, elevating marginal bone levels with increased KM, and minimizing marginal bone loss with thicker mucosal tissue [25,26]. In addition, palatal soft tissue is the site where OLL occurs least frequently, and the graft contains an appropriate amount of genetically predetermined lamina propria overlying the epithelium, which maintains its color and shape at the grafted site, regardless of the transplantation site [27]. Therefore, the recipient site of the excised OLL could be expected to be transformed into tissues with characteristics of palatal tissue. Since the risk of recurrence still exists and the removal of the suspected causative factor of OLL is not possible, periodic recall checks should be maintained, even though the clinical outcomes after FGG have been shown to be reliable.

A limitation of the present case report is that it provides limited evidence-based information to clinicians as this was a single-case study. Still, a rare case of OLL associated with implant prosthetic delivery occurring in multiple areas within a patient is introduced in the present case report, and it shows that augmentation of KM collected from palatal soft tissue can alter peri-implant soft tissue to a healthier condition with remission of OLL for long-term 10-year follow-up. If additional research is conducted in the future, it is anticipated that reasonable guidelines regarding the management of OLL can be proposed.

## 4. Conclusions

Within the limitations of this case report, it is suggested that dental implants or prostheses could be causative factors for OLL, and the surgical approach of FGG with palatal tissue was proven to be effective in a case that did not respond to conservative treatment.

## Figures and Tables

**Figure 1 medicina-59-02188-f001:**
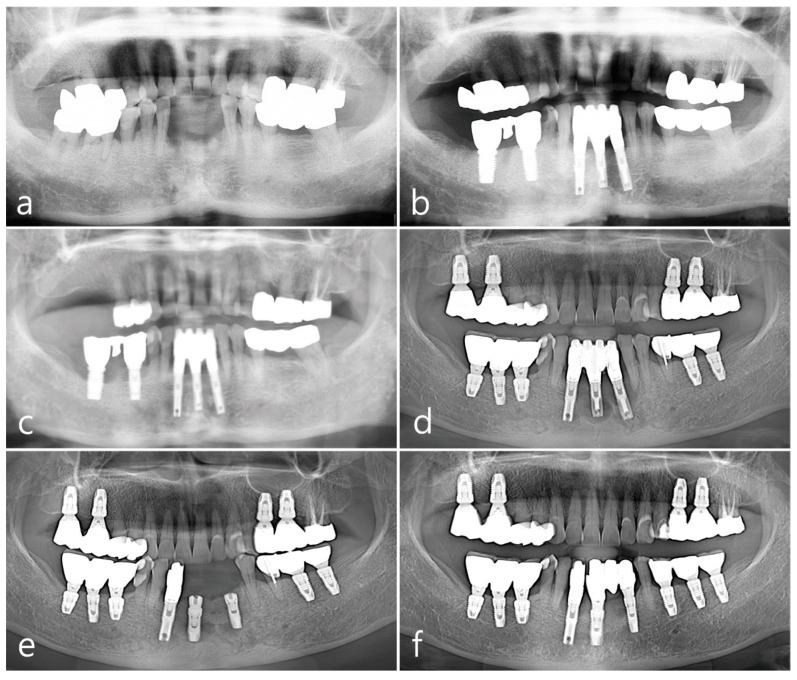
Panoramic views during the 21-year follow-up period. Radiograph taken in 2000 for baseline visit (**a**), in 2002 (**b**), in 2011 (**c**), in 2014 (**d**), in 2019 (**e**), and in 2021 (**f**).

**Figure 2 medicina-59-02188-f002:**
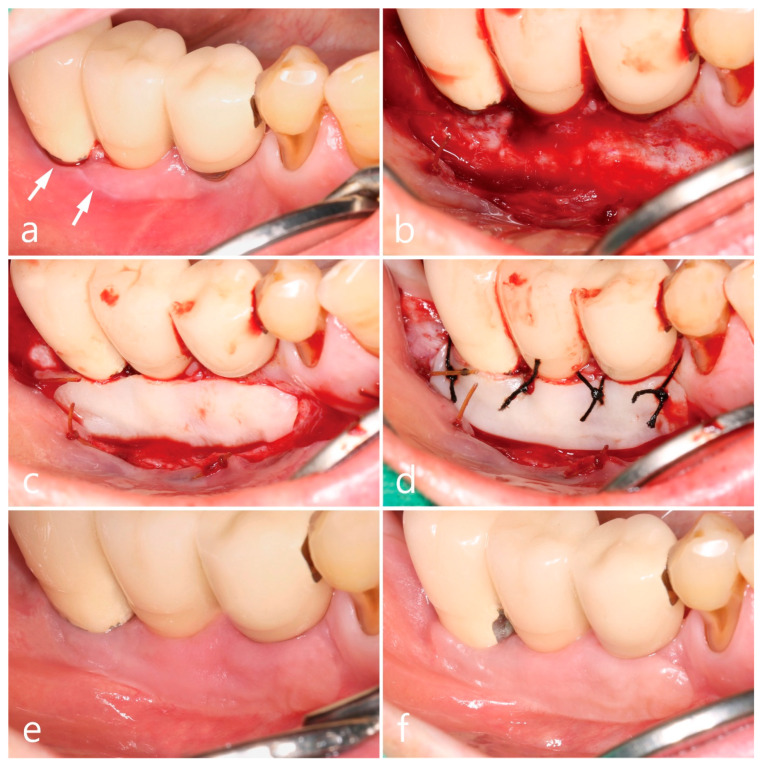
Clinical aspects in the mandibular right posterior region. Oral lichenoid lesion with white striae (white arrows) (**a**), free gingival graft procedures (**b**–**d**), and postoperative follow-up for 3 months (**e**) and 10 years (**f**).

**Figure 3 medicina-59-02188-f003:**
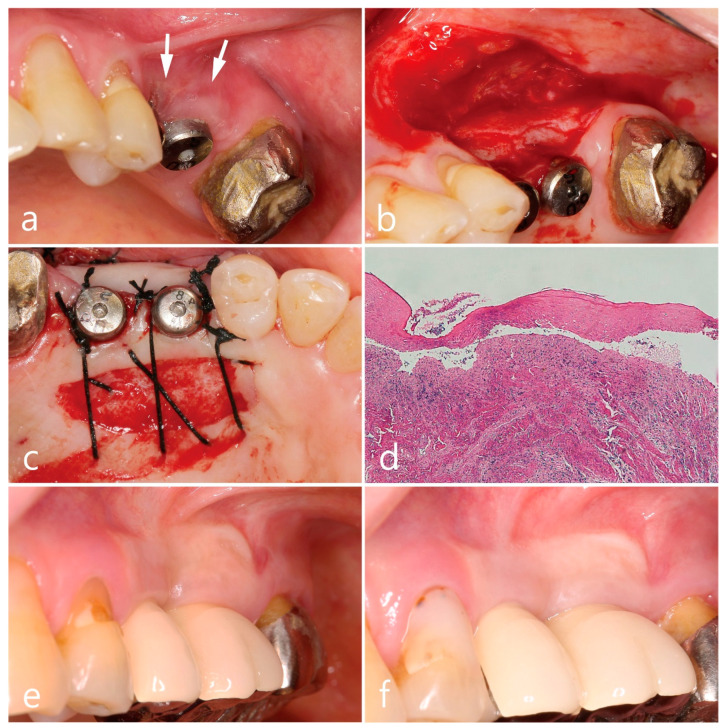
Clinical aspects in the maxillary left posterior region. Wickham’s striae and epithelial desquamation (white arrow) after 4 months of implant placement (**a**), free gingival graft procedures (**b**,**c**), histologic feature of excised soft tissue specimen (**d**), and postoperative follow-up for 6 months (**e**) and 9 years (**f**).

**Figure 4 medicina-59-02188-f004:**
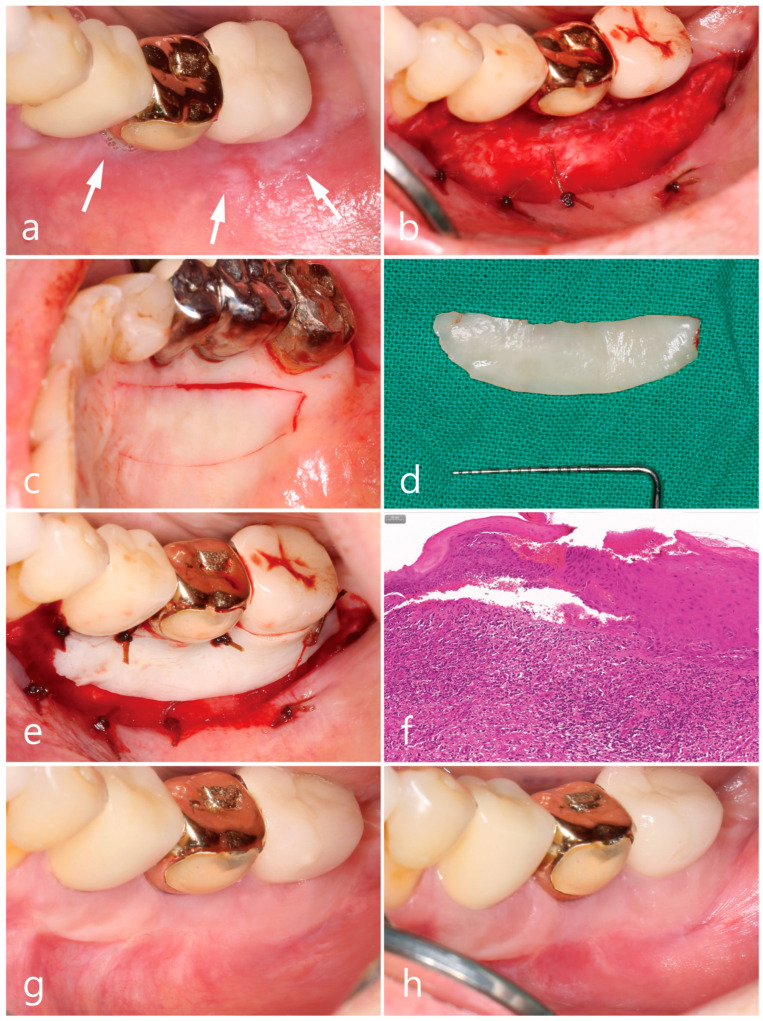
Clinical aspects in the mandibular left posterior region. Oral lichenoid lesion with white striae (white arrows) (**a**), free gingival graft procedures (**b**–**e**), histologic feature of excised soft tissue specimen (**f**), and postoperative follow-up for 6 months (**g**) and 12 months (**h**).

## Data Availability

Data are contained within the article.

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
