# Peer review of "Oral Lichenoid Lesion following Dental Implant Placement and Successful Management with Free Gingival Graft: A Case Report with 10-Year Follow-Up"

_medicina, 2023, doi:10.3390/medicina59122188_

Round 1

Reviewer 1 Report

Comments and Suggestions for Authors

Dear Editor

I would like to mention a couple points to improve the contents:

-          Please provide information on the type of prosthetic restoration.

-          Please list the medications employed for palliative treatment.

-          Was a biopsy and histopathological examination not conducted for lesions in the mandibular right posterior region?

-          Why hasn't there been an effort to alter the prosthetic restoration in the context of oral lichenoid lesion, given the crucial significance of removing or changing the local trigger in its management?

-          Line 90: How was the oral hygiene condition of the patient during the 2011 visit?

-          Line 184: The reference number is not correct.

-          Figure 2, Caption: two point (..) is placed at the end.

Comments on the Quality of English Language

-

Author Response

  1. Please provide information on the type of prosthetic restoration.

Answer: Thank you for the comment. We added the information on the type of prosthetic restoration in the manuscript as follows.

2.1. Mandibular right posterior region

(L106) and reconstructed as cement-retained porcelain-fused-metal (PFM) three-unit bridge with

(L129-130) were placed and cement-retained PFM splinted crowns were delivered (Figure 1d),

2.2. Maxillary left posterior region

(L157-158 ) The second premolar and first molar in the left maxilla were placed with two SLA surface implants (Implantium®) and cement-retained PFM splinted crowns were delivered in 2013.

2.3. Mandibular left posterior region

(L184-186 ) Cement-retained resin-faced single gold crown was delivered at first molar implant, and PFM single crown was made for each second molar and premolar implant.

  1. Please list the medications employed for palliative treatment.

Answer: Thank you for the comment. We added the medications as follows.

(L113-115 ) For palliative therapy, topical corticosteroid 0.1% dexamethasone ointment (Peridex, Green Cross Corp., Yongin, Korea) was repeatedly administered and recall visits were made every 2~3 weeks for 4 months; however, the improvement in symptoms was insignificant,

(L160-162 ) Oral steroid prednisolone (Solondo 5mg) was administered 3 times a day for the first 2 weeks and a mouth rinse of 10 mL of 0.05% dexamethasone gargle was used concomitantly.

  1. Was a biopsy and histopathological examination not conducted for lesions in the mandibular right posterior region?

Answer: Thank you for the comment. OLL at mandibular right posterior region which we encountered first among others were diagnosed under clinical criteria as it showed quite typical reticular type of OLP. Recurrence after second implantation at 2011 also showed similar clinical aspects and symptoms with previous ones. However, as the lesions were multiply occurred at different sites, we decided to perform further histopathologic assessment with surgical excisions to exclude dysplasia or malignancy.

  1. Why hasn't there been an effort to alter the prosthetic restoration in the context of oral lichenoid lesion, given the crucial significance of removing or changing the local trigger in its management?

Answer: Thank you for the important comments. We totally agree that removal or replacement of dental materials is crucial when oral lichenoid contact lesions are associated. We also have added some clinical reports on complete or partial resolution of OLL following replacement of amalgam restorations, and 71% of patients showed regression after removal or replacement of dental restorative materials including amalgam and other metals including gold, nickel, and so on (Introduction L50-53; L77-85). Despite the previous results, there still remains unpredictability on the complete resolution, risks of potential iatrogenic damage during the dental treatment and possible allergic reactions to new restoration materials as describe in the discussion (Discussion L237-245). It was not guaranteed that which materials among the metal alloys were responsible for the lesion and the patient showed OLL at both PFM and gold crown which limited the alternative choices for the replacement. In addition, the implants were surrounded by movable non-keratinized tissue which made patient’s oral hygiene performance more difficult and susceptible to the inflammatory response. Augmentation of KM was finally chosen to excise OLL and to correct peri-implant tissue condition for oral health improvement as well as resolution of OLL. 

  1. Line 90: How was the oral hygiene condition of the patient during the 2011 visit?

Answer: Thank you for the comment. Patient attended to the recall visit irregularly with the intervals of 16 ~ 24 months after the first FGG as she felt no specific complaints. Oral hygiene performance in interdental cleaning at posterior teeth was not adequate and showed plaque accumulation with slight gingival redness and bleeding on probing. We added the description about the oral hygiene condition as follows.

(L120-126) Patient’s attendance to the maintenance visit after the FGG was irregular to show 16 ~ 24 months of recall intervals. Self-performed mechanical plaque control in the inter-proximal areas of posterior teeth was not adequately done and easily showed bleeding on probing with plaque accumulation and gingival redness. At every visit, thorough whole mouth scaling and root planing with ultrasonic scaler (EMS, Nyon, Switzerland) and hand instrument was performed.

  1. Line 184: The reference number is not correct.

Answer: Thank you for the comment. A comma (,) was missing between reference 7 and 10. We corrected [710] to [12,21] as more references were added in the manuscript.

  1. Figure 2, Caption: two point (..) is placed at the end.

Answer: Thank you for the comment. We removed one of the periods.

Reviewer 2 Report

Comments and Suggestions for Authors

Dear Authors,

Thank you for the opportunity to review you paper, here attached you can find my suggestions.

Regards

Comments on the Quality of English Language

Minor editing of English language required

Author Response

Introduction

  1. The etiological and histopathological distinction between Lichen Planus and Lichenoid Lesions is not very clear, it would be correct for a non-expert reader to understand the difference between the two pathologies and understand that they have a different clinical course. Lichen Planus is a chronic potentially malignant lesion where the affected patient must be subjected to traditional implant surgery with caution, carefully evaluating whether there are degrees of dysplasia or pathology with active inflammation before performing surgery. Lichenoid lesions, on the other hand, have the particularity of reverting when the triggering factor is eliminated or treated with specific protocols.

Answer: Thank you for the important comment. For more information, we added some contents about potential malignancy of OLP (Introduction, L 42-44), necessity in periodic monitoring of OLP for potential malignancy and control of active lesion (L59-64), and resolution of OLL after removal of dental materials (L50-52, L80-83) with new references. It was described as follows.

(L42-44) Although the malignant potential of OLP is controversial, a few large retrospective studies have reported a higher risk of malignant transformation to squamous cell car-cinoma (SCC) in OLP populations [4,5].

Reference:

  1. Rodstrom, P.O., Jontell, M., Mattsson, U., Holmberg, E. Cancer and oral lichen planus in a Swedish population. Oral Oncol. 2004, 40, 131-138.
  2. Gandolfo, S., Richiardi, L., Carrozzo, M., Broccoletti, R., Carbone, M., Pagano, M., Vestita, C., Rosso, S., Merletti, F. Risk of oral squamous cell carcinoma in 402 patients with oral lichen planus: a follow-up study in an Italian population. Oral Oncol. 2004, 40, 77-83.

(L59-64) A prospective study reported high failure of 42 implants out of 55 within short loading period of 7 to 11 weeks in active erosive phase, which suggested that implant surgery should be avoided until the remission of atrophic or erosive lesion was achieved [11]. In addition, it is necessary to exclude dysplasia and SCC by histopathologic assessment before the active treatment and periodically monitor the possible malignant transfor-mation of OLP.

Reference:

  1. Aboushelib, M.N., Elsafi, M.H. Clinical management protocol for dental implants inserted in patients with active lichen planus. J. Prosthodont. 2017, 26, 29–33.

(L50-53) Oral lichenoid contact reactions are most commonly associated with dental amalgam and some studies have reported complete or partial resolution of oral lichenoid contact lesion following the replacement of amalgam restorations with positive patch test [8,9].

Reference:

  1. Dunsche, A., Kastel, I., Tenheyden, H., Springer, I.N.G., Christopers, E., Brash, J. Contact dermatitis and allergy; oral lichenoid reactions associated with amalgam; improvement after amalgam removal. Br. J. Dermatology. 2003, 148, 70-76.
  2. Suter, V.G.A., Warnakulasuriya, S. The role of patch testing in the management of oral lichenoid reactions. J. Oral Pathol. Med. 2016, 45, 48-57.

(L80-83) Patients with lichenoid contact reactions reported to have higher prevalence of dental metal allergy and 71% of patients showed regression after removal or replacement of dental restorative materials [16]. In patch-tested subjects, the most frequent allergens were mercury followed by gold and nickel.

Reference:

  1. Mårell, L., Tillberg, A., Widman, L., Bergdahl, J., Berglund, A. Regression of oral lichenoid lesions after replacement of dental restorations. J. Oral Rehabil. 2014, 41, 381-391.

  1. It is true that the studies conducted on the survival of implants in patients affected by Lichen Planus have resulted in success comparable to healthy patients, but it is necessary to distinguish whether there are studies that investigate implant success after the onset of lichenoid lesions and whether implant materials could be the trigger for such lesions, or whether the cause of the appearance of the lesions is mainly due to the prosthesis and not to the implant itself.

Answer: Thank you for the important comment. Most of the studies were dealing with the dental implants placed in the patients who already showed OLP, and controlled before the surgery, which resulted in favorable survival rates comparable to the healthy patient. In case of oral lichenoid contact lesions associated with dental restorations, there were limited numbers of studies on OLL with amalgam restoration and complete or partial resolution after the removal (Introduction L50-53). About the OLL associated with dental implant and its restoration, a few clinical studies reported allergic reaction of titanium, however, the clinical features of the lesions were mostly granulomatous rather than lichen planus-like appearance. Another clinical report on the dental restorative materials including amalgam, gold, nickel, etc. associated with OLL was added in the manuscript, which showed resolution of the lesion after the removal (Introduction L70-85). These were described as follows.

(L50-53) Oral lichenoid contact reactions are most commonly associated with dental amalgam and some studies have reported complete or partial resolution of oral lichenoid contact lesion following the replacement of amalgam restorations with positive patch test [8,9].

Reference:

  1. Dunsche, A., Kastel, I., Tenheyden, H., Springer, I.N.G., Christopers, E., Brash, J. Contact dermatitis and allergy; oral lichenoid reactions associated with amalgam; improvement after amalgam removal. Br. J. Dermatology. 2003, 148, 70-76.
  2. Suter, V.G.A., Warnakulasuriya, S. The role of patch testing in the management of oral lichenoid reactions. J. Oral Pathol. Med. 2016, 45, 48-57.

(L70-85) A clinical study reported titanium (Ti) allergy with the low prevalence of 0.6% from 1500 implant patients, who showed clinical features including allergic symptoms after implant surgery, de-keratinized hyperplastic lesions of peri-implant soft tissues, unexplained implant failures such as spontaneous rapid exfoliation, history of multiple allergies, etc [13]. However, the lesion is frequently granulomatous on the soft tissues and persistent pain from the bone may present, which are rather distinctive from typical oral lichenoid contact lesion. Still, the significance of Ti as a cause of allergic reactions in dental implants are not proven [14]. Another retrospective study investigated the proportions and characteristics of patients who complained of pain and discomfort in oral mucosal lesions following dental treatment, OLP and allergic reaction occurred most commonly after implant treatment among patients [15]. Patients with lichenoid contact reactions reported to have higher prevalence of dental metal allergy and 71% of patients showed regression after removal or replacement of dental restorative materials [16]. In patch-tested subjects, the most frequent allergens were mercury followed by gold and nickel. Previous studies have limitations that the sample size is very small, heterogenous dental materials are included and cannot clearly identify the causal agent of the lesions.

Reference:

  1. Sicilia, A., Cuesta, S., Coma, G., Arregui, I., Guisasola, C., Ruiz, E., Maestro, A. Titanium allergy in dental implant patients: a clinical study on 1500 consecutive patients. Clin. Oral. Implants. Res. 2008, 19, 823–835.
  2. Javed, F., Al-Hezaimi, K., Almas, K., Romanos, G.E. Is titanium sensitivity associated with allergic reactions in patients with dental implants? A systematic review. Clin. Implant. Dent. Relat. Res. 2013, 15, 47-52.
  3. Ju, H., Ahn, Y., Jeong, S., Jeon, H., Kim, K., Song, B., Ok, S. Charateristics of patients who preceive dental treatment as a cause of oral mucosal lesions. J. Oral. Sci. 2019, 61, 468-474.
  4. Mårell, L., Tillberg, A., Widman, L., Bergdahl, J., Berglund, A. Regression of oral lichenoid lesions after replacement of dental restorations. J. Oral Rehabil. 2014, 41, 381-391.

Case Presentation:

  1. Therapeutic alternatives regarding the treatment of lichenoid lesions should be indicated. It is known that there are non-surgical alternatives and the choice of FGG should be justified.

Answer: Thank you for the comment. Despite the topical and systemic corticosteroid therapy, improvement of the symptoms in the OLL was not achieved. Removal or replacement of dental materials is crucial when oral lichenoid contact lesions are associated. We also have added some clinical reports on complete or partial resolution of OLL following replacement of amalgam restorations, and 71% of patients showed regression after removal or replacement of dental restorative materials including amalgam and other metals including gold, nickel, and so on (Introduction L50-53; L77-85). Despite the previous results, there still remains unpredictability on the complete resolution, risks of potential iatrogenic damage during the dental treatment and possible allergic reactions to new restoration materials as describe in the discussion (Discussion L237-245). It was not guaranteed that which materials among the metal alloys were responsible for the lesion and the patient showed OLL at both PFM and gold crown which limited the alternative choices for the replacement. In addition, the implants were surrounded by movable non-keratinized tissue which made patient’s oral hygiene performance more difficult and susceptible to the inflammatory response. Augmentation of KM was finally chosen to excise OLL and to correct peri-implant tissue condition for oral health improvement as well as resolution of OLL. In the case presentation section we have added the reason to choose FGG as follows.

(L112-118) For palliative therapy, topical corticosteroid 0.1% dexamethasone ointment (Peridex, Green Cross Corp., Yongin, Korea) was repeatedly administered and recall visits were made every 2~3 weeks for 4 months; however, the improvement in symptoms was insignificant, and the lesion recurred continuously. In addition, the mandibular right posterior region had shallow vestibule and lack of keratinized mucosa around implant prosthesis, which made the patient’s cleansing performance difficult. Therefore, a surgical approach involving keratinized tissue augmentation using FGG was performed, which resulted in the disappearance of OLL until the visit in 2011 (8 years after the first FGG).

(L162-166) The painful lesion persisted despite the topical and systemic corticosteroid therapy. In addition, high frenal attachment and movable mucosal margin around dental implants facilitated plaque accumulation and gingival inflammation. To correct the peri-implant mucosal anatomic conditions and excision of OLL, surgical intervention using FGG was performed.

(L190-193) Similar to the other two regions described above, mandibular posterior region showed no specific improvement in symptoms despite topical and systemic corticosteroid therapies, and patient felt difficulty in brushing around the implant due to the movable non-keratinized mucosa.

  1. What parameters were established to evaluate the absence of pathology and the health of the tissues?

Answer: Thank you for the comment. Parameters for the evaluation on the tissue health state after the FGG could be referred from the article by Axéll and Henriksen (2007) who presented 4-point clinical grade scale (0-3) for healing scores on the free palatal grafts to treat gingival lichen planus/lichenoid lesions. It was added to the manuscript as follows.

(L139-147) To evaluate the pathologic tissue condition after the healing of FGG, intraoral clinical photograph was taken under standardized set and a scoring system with 4-point clinical grade scale (0-3) by Axéll and Henriksen# was applied. In short, it was subjectively assessed as grade 0 for no improvement or aggravation, grade 1 for improvement but with extensive erythema and/or symptoms, grade 2 for improvement but with some erythema and no symptoms and grade 3 for healing with neither erythema nor symptoms. Healing score in the mandibular right region appeared grade 3 and a healthy mucosal condition was maintained for up to 10 years of follow-up (Figure 2f).

(L174) Healing score was grade 3 and there was no recurrence of the lesion observed at 6 months (Figure 3e) or 9 years of follow-up (Figure 3f) after prosthetic loading.

(L201) Healing score was grade 3 and increased width of the keratinized mucosa in healthy condition around the implants was shown at 6 and 12 months of follow-up (Figure 4h, h).

Reference:

  1. Axéll, T.; Henriksen, B.M. Treatment of gingival lichen with free palatal grafts. J. Oral. Pathol. Med. 2007, 36, 105-109.

  1. Has the patient undergone periodic professional oral hygiene sessions? If so, what types of treatment have been conducted to maintain oral and periodontal health?

Answer: Thank you for the comment. Patient attended to the recall visit irregularly with the intervals of 16 ~ 24 months after the first FGG and until year 2011 as she felt no specific complaints. Oral hygiene performance in interdental cleaning at posterior teeth was not adequate and showed plaque accumulation with slight gingival redness and bleeding on probing. After the explantation and replacement of dental implants done in right mandibular lesion, the patient’s recall intervals were shortened within a year although the attendance was still irregular. Thorough whole mouth scaling and root planing was performed at every visit. It was described in the manuscript as follows.

(L120-126) Patient’s attendance to the maintenance visit after the FGG was irregular to show 16 ~ 24 months of recall intervals. Self-performed mechanical plaque control in the interproximal areas of posterior teeth was not adequately done and easily showed bleeding on probing with plaque accumulation and gingival redness. At every visit, thorough whole mouth scaling and root planing with ultrasonic scaler (EMS, Nyon, Switzerland) and hand instrument was performed.

(L147-149) Although the patient still showed irregular compliance for the recall visit, the interval was shortened within a year and professional mechanical plaque control was repeated for supportive maintenance therapy.

Reviewer 3 Report

Comments and Suggestions for Authors

Line 43-44-45: Please insert some references about what you are writing.

Line 84-89 and 95-100: Regarding the mandibular right posterior region, did you ever perform a histological examination of the suspected lesion in 2002 and 8 years later? According to what was made a diagnosis of OLL?

Just for sharing: In these cases, we prescribe daily washing with water and sodium bicarbonate and corticosteroids only in the acute phases

Author Response

  1. Line 43-44-45: Please insert some references about what you are writing.

Answer: Thank you for the comment. We added reference [6] in L47.

  1. Line 84-89 and 95-100: Regarding the mandibular right posterior region, did you ever perform a histological examination of the suspected lesion in 2002 and 8 years later? According to what was made a diagnosis of OLL?

Answer: Thank you for the comment. OLL at mandibular right posterior region which we encountered first among others were diagnosed under clinical criteria as it showed quite typical reticular type of OLP. Recurrence after second implantation at 2011 also showed similar clinical aspects and symptoms with previous ones. However, as the lesions were multiply occurred at different sites, we decided to perform further histopathologic assessment with surgical excisions to exclude dysplasia or malignancy.

Reviewer 4 Report

Comments and Suggestions for Authors

Introduction line 58-59 (Cases of OLL developing after implant or prosthesis placement have rarely been reported). It is important to mention approximately how many cases have been published.

Case Presentation: In the line 65. The authors not mentioned if the patient received private care or dental service at a University.

In the line 208: A limitation of the present case report is that it provides limited evidence-based in-208 formation to clinicians as this was a single-case study.  It is necessary to support the idea that only one clinical case was reported and what advantage was obtained from the 10-year follow-up.

Comments on the Quality of English Language

The quality of the language is good

Author Response

  1. Introduction line 58-59 (Cases of OLL developing after implant or prosthesis placement have rarely been reported). It is important to mention approximately how many cases have been published.

Answer: Thank you for your comment. Most of the studies were dealing with the dental implants placed in the patients who already showed OLP, and controlled before the surgery, which resulted in favorable survival rates comparable to the healthy patient. In case of oral lichenoid contact lesions associated with dental restorations, there were limited numbers of studies on OLL with amalgam restoration and complete or partial resolution after the removal (Introduction L50-53). About the OLL associated with dental implant and its restoration, a few clinical studies reported allergic reaction of titanium, however, the clinical features of the lesions were mostly granulomatous rather than lichen planus-like appearance. Another clinical report on the dental restorative materials including amalgam, gold, nickel, etc. associated with OLL was added in the manuscript, which showed resolution of the lesion after the removal (Introduction L70-85). These were described as follows.

(L50-53) Oral lichenoid contact reactions are most commonly associated with dental amalgam and some studies have reported complete or partial resolution of oral lichenoid contact lesion following the replacement of amalgam restorations with positive patch test [8,9].

Reference:

  1. Dunsche, A., Kastel, I., Tenheyden, H., Springer, I.N.G., Christopers, E., Brash, J. Contact dermatitis and allergy; oral lichenoid reactions associated with amalgam; improvement after amalgam removal. Br. J. Dermatology. 2003, 148, 70-76.
  2. Suter, V.G.A., Warnakulasuriya, S. The role of patch testing in the management of oral lichenoid reactions. J. Oral Pathol. Med. 2016, 45, 48-57.

(L70-85) A clinical study reported titanium (Ti) allergy with the low prevalence of 0.6% from 1500 implant patients, who showed clinical features including allergic symptoms after implant surgery, de-keratinized hyperplastic lesions of peri-implant soft tissues, unexplained implant failures such as spontaneous rapid exfoliation, history of multiple allergies, etc [13]. However, the lesion is frequently granulomatous on the soft tissues and persistent pain from the bone may present, which are rather distinctive from typical oral lichenoid contact lesion. Still, the significance of Ti as a cause of allergic reactions in dental implants are not proven [14]. Another retrospective study investigated the proportions and characteristics of patients who complained of pain and discomfort in oral mucosal lesions following dental treatment, OLP and allergic reaction occurred most commonly after implant treatment among patients [15]. Patients with lichenoid contact reactions reported to have higher prevalence of dental metal allergy and 71% of patients showed regression after removal or replacement of dental restorative materials [16]. In patch-tested subjects, the most frequent allergens were mercury followed by gold and nickel. Previous studies have limitations that the sample size is very small, heterogenous dental materials are included and cannot clearly identify the causal agent of the lesions.

Reference:

  1. Sicilia, A., Cuesta, S., Coma, G., Arregui, I., Guisasola, C., Ruiz, E., Maestro, A. Titanium allergy in dental implant patients: a clinical study on 1500 consecutive patients. Clin. Oral. Implants. Res. 2008, 19, 823–835.
  2. Javed, F., Al-Hezaimi, K., Almas, K., Romanos, G.E. Is titanium sensitivity associated with allergic reactions in patients with dental implants? A systematic review. Clin. Implant. Dent. Relat. Res. 2013, 15, 47-52.
  3. Ju, H., Ahn, Y., Jeong, S., Jeon, H., Kim, K., Song, B., Ok, S. Charateristics of patients who preceive dental treatment as a cause of oral mucosal lesions. J. Oral. Sci. 2019, 61, 468-474.
  4. Mårell, L., Tillberg, A., Widman, L., Bergdahl, J., Berglund, A. Regression of oral lichenoid lesions after replacement of dental restorations. J. Oral Rehabil. 2014, 41, 381-391.

  1. Case Presentation: In the line 65. The authors not mentioned if the patient received private care or dental service at a University.

Answer: Thank you for the comment. Implant treatment and soft tissue surgery (FGG) were done at a private practice in periodontics and implant dentistry. We added this information in line 65-66 as follows.

A 45-year-old female patient who visited a private practice in periodontics and implant dentistry underwent dental implant treatment following the extraction of teeth

  1. In the line 208: A limitation of the present case report is that it provides limited evidence-based information to clinicians as this was a single-case study.  It is necessary to support the idea that only one clinical case was reported and what advantage was obtained from the 10-year follow-up.

Answer: Thank you for the comment. We have added some advantageous aspects of the present case report as follows.

(L272-277) A limitation of the present case report is that it provides limited evidence-based information to clinicians as this was a single-case study. Still, a rare case of OLL associated with implant prosthetic delivery occurred in multiple areas within a patient is introduced in the present case report and it shows that augmentation of KM collected from palatal soft tissue can alter peri-implant soft tissue to a healthier condition with remission of OLL for long-term 10-year follow-up.

Round 2

Reviewer 1 Report

Comments and Suggestions for Authors

This paper can be accepted for publication.